# Association of part-time clinical work with well-being and mental health in General Internal Medicine: A survey among Swiss hospitalists

Lisa Bretagne[1]*, Stefanie Mosimann[1,2], Christine Roten[1], Martin Perrig[1], Daniel Genné[3], Manfred Essig[4], Marco Mancinetti[5], Marie Méan[6], Pauline Darbellay Farhoumand[7], Lars C. Huber[8], Elisabeth Weber[9], Christoph Knoblauch[10], Andreas W. Schoenenberger[11], Sonia Frick[12], Eliane Wenemoser[13], Daniel Ernst[14], Michael Bodmer[15], Drahomir Aujesky[1], Christine Baumgartner[1]

1 Department of General Internal Medicine, Inselspital, Bern University Hospital, University of Bern, Bern, Switzerland, 2 School of Health Professions Education, Maastricht University, Maastricht, The Netherlands, 3 Department of Internal Medicine, Hospital of Biel-Bienne, Bienne, Switzerland, 4 Department of General Internal Medicine, Tiefenau Hospital, Bern, Switzerland, 5 Department of Medicine, University and Hospital of Fribourg, Fribourg, Switzerland, 6 Department of Internal Medicine, Centre Hospitalier Universitaire Vaudois (CHUV), University of Lausanne, Lausanne, Switzerland, 7 Department of Medicine, Geneva University Hospitals (HUG), Geneva, Switzerland, 8 Department of General Internal Medicine, Stadtspital Zürich Triemli, Zürich, Switzerland, 9 Department of General Internal Medicine, Stadtspital Zürich Waid, Zürich, Switzerland, 10 Department of General Internal Medicine, Hospital of Nidwalden, Stans, Switzerland, 11 Department of General Internal Medicine, Cantonal Hospital of Münsterlingen, Münsterlingen, Switzerland, 12 Department of General Internal Medicine, Hospital of Limmattal, Schlieren, Switzerland, 13 Department of General Internal Medicine, Hospital Region of Oberaargau, Langenthal, Switzerland, 14 Department of General Internal Medicine, Hospital of Thun, Thun, Switzerland, 15 Department of General Internal Medicine, Cantonal Hospital of Zug, Baar, Switzerland

* lisa.bretagne@gmail.com

## Abstract

## Introduction

Burnout and low job satisfaction are increasing among the General Internal Medicine (GIM) workforce. Whether part-time compared to full-time clinical employment is associated with better wellbeing, job satisfaction and health among hospitalists remains unclear.

## Materials and methods

We conducted an anonymized cross-sectional survey among board-certified general internists (i.e. hospitalists) from GIM departments in 14 Swiss hospitals. Part-time clinical work was defined as employment of <100% as a clinician. The primary outcome was well-being, as measured by the extended Physician Well-Being Index (ePWBI), an ePWBI ≥3 indicating poor wellbeing. Secondary outcomes included depressive symptoms, mental and physical health, and job satisfaction. We compared outcomes in part-time and full time workers using propensity score-adjusted multivariate regression models.

**Data Availability Statement:** Data cannot be shared publicly to preserve the anonymity of participants. As the sample size is relatively small,

some participants may be recognized due to the gathered personal information. The data is available upon request from the first or last authors (Lisa Bretagne and Christine Baumgartner), or from the Department of General Internal Medicine at the Inselspital, Bern (forschung.kaim@insel.ch).

**Funding:** This work was supported by a grant from the Swiss Society of General Internal Medicine (SGAIM) Foundation (https://www.sgaim.ch/de/themen/forschung/portraet-sgaim-foundation) awarded to L.B. The funders had no role in study design, data collection and analysis, decision to publish, or preparation of the manuscript.

**Competing interests:** The authors have declared that no competing interests exist.

## Results

Of 199 hospitalists invited, 137 (69%) responded to the survey, and 124 were eligible for analysis (57 full-time and 67 part-time clinicians). Full-time clinicians were more likely to have poor wellbeing compared to part-time clinicians (ePWBI ≥3 54% vs. 31%, p = 0.012). Part-time compared to full-time clinical work was associated with a lower risk of poor well-being in adjusted analyses (odds ratio 0.20, 95% confidence interval 0.07–0.59, p = 0.004). Compared to full-time clinicians, there were fewer depressive symptoms (3% vs. 18%, p = 0.006), and mental health was better (mean SF-8 Mental Component Summary score 47.2 vs. 43.2, p = 0.028) in part-time clinicians, without significant differences in physical health and job satisfaction.

## Conclusions

Full-time clinical hospitalists in GIM have a high risk of poor well-being. Part-time compared to full-time clinical work is associated with better well-being and mental health, and fewer depressive symptoms.

## Introduction

Physicians' well-being has been shown to positively contribute to patient satisfaction and quality of care [1]. In recent years, studies have shown a decrease in physician well-being and career satisfaction, as well as an increase in burnout [2, 3]. General internal medicine (GIM) is particularly affected by this trend [4, 5]. A recent study among GIM residents in Switzerland showed that 19% had a low well-being and 60% felt burnt-out [6]. More than a fifth of the residents in this study regretted their career choice [6]. Combined with an expected increase in the need for GIM physicians both in the ambulatory and hospital setting due to the ageing population and the growing number of complex and multimorbid patients, changes are needed in career models for GIM to maintain enough board-certified general internists caring for hospitalized patients (hereinafter referred to as hospitalists) in the years to come [7].

Several studies have investigated causes of burnout and factors that influence well-being [3]. Having control over one's schedule and the number of hours worked weekly have been identified as important predictors of both a better work-life balance and better career satisfaction [8]. The Swiss Medical Association (FMH) has suggested that promotion of part-time work may increase retention of physicians in clinical functions [9]. Another study has shown that the opportunity to engage in other activities than clinical work that were deemed meaningful by participants, such as research or teaching, was associated with lower rates of burnout [10]. In order to prevent migration of general internists from the hospital setting to ambulatory care, experts have called for increasing opportunities for part-time work in the GIM hospital setting [7]. However, the impact of working part-time as a clinician in a hospital setting on physician well-being, job satisfaction, and health has not been sufficiently investigated. Previous studies in the USA, the Netherlands, or Germany suggested an increase in job satisfaction and decrease in stress and burnout with part-time compared to full-time work among physicians, but results on work-life balance were conflicting [11–16]. Those studies were conducted either in specialties such as pediatrics, radiology, or gynecology, or did not take into account their specialty. One study conducted in a hospital setting in GIM found no difference in overall job satisfaction between part-time and full-time physicians, however, well-being and

health were not assessed [17]. In addition, these studies did not discriminate between part-time work overall and part-time clinical work with protected time for non-clinical tasks, which may contribute to improved wellbeing without reducing overall working hours.

The aim of this study was to assess the association between part-time compared to full-time clinical work and aspects of well-being, job satisfaction, and health among hospitalists of GIM departments in Switzerland.

## Materials and methods

### Study design and population

We conducted an anonymized cross-sectional electronic survey among hospitalists in Switzerland. The Swiss health care system is characterized by universal access that is ensured through mandatory health care insurance [18]. Among practicing physicians, 22% are specialized in GIM, making it the most frequent specialization in Switzerland [19]. Board certification for GIM in Switzerland requires a 5-year training; [20] GIM residents can then transition to a position either as a hospitalist or an outpatient general practitioner, or may still decide to pursue a training in another medical specialty, as many subspecialties (e.g., cardiology, infectious diseases, etc.) require at least 2 years of training in GIM. Patients hospitalized on a GIM ward of a Swiss teaching hospital are cared for by a GIM resident and a supervising (board-certified) hospitalist, and other specialists are involved depending on the patient's disease and on request of the hospitalist. The hospitalists themselves are supervised regularly (e.g., once a week) or upon request by a chief physician, who is also responsible for running the department and for other administrative tasks, teaching, and research. Overall, the hospitalists have the overview and bear the ultimate responsibility for optimal patient care.

The survey was sent to hospitalists from GIM departments of 14 Swiss hospitals, including small regional hospitals and large tertiary care centres from various Swiss regions. We deliberately excluded resident physicians from the survey as a considerable part of the residents working in GIM departments will switch to speciality training [21]. Furthermore, few hospitals in Switzerland allow part-time employment of residents in GIM, thus only few residents work part-time in GIM, [6] and it would have been difficult to get a big enough sample size of participants for meaningful results. By focusing on hospitalists, we intended for the study population to be more homogeneous and representative of the "general internist". We excluded chief-physicians in hospital medicine, hospitalists without direct patient care on a ward, those who had newly transitioned to a position as hospitalist within the last month or those who intended to leave GIM within the next month, as their answers may not be representative of the GIM hospitalist workforce. The aim of our study and use of data for publication was stated at the beginning of our survey, and we indicated that participation in the survey would be taken as consent. Because the survey was anonymized, we were unable to collect individual informed consent forms. The authors had no access to information that could identify individual participants. This study has been granted an exemption from requiring ethical approval due to its nature.

### Data collection

The survey was conducted using the public survey function of REDCap (Research Electronic Data Capture), which allows collection of anonymized responses. Study data were collected and managed using REDCap electronic data capture tools hosted at the CTU, Bern [22, 23]. The survey consisted of questions to assess the physicians' demographic data, work percentage and schedule, and measures of well-being, job satisfaction and health. The survey link was sent

to the hospitalists by the project collaborators of each participating hospital via e-mail in December 2021. A reminder was sent per email after 1 week, 2 weeks and 4 weeks.

For this study, full-time clinical work was defined as a full time (i.e. 100%) employment as a clinical hospitalist, which corresponds to 46–50 clinical hours per week in Switzerland. Part-time clinical work was defined as level of employment <100% for clinical work (i.e. less than 46 clinical hours per week). Physicians employed full time, but with a predefined percentage of their work specifically dedicated to non-clinical tasks (i.e. research, teaching, or administration), were considered as part-time clinicians, because we hypothesized that diversification of one's work and engaging in meaningful non-clinical tasks may affect well-being similar to engaging in meaningful tasks outside of the hospital during time off [10]. The percentage of clinical employment was based on self-report of the number of hours specified in hospitalists' contracts, and did not include overtime.

## Outcomes

The primary outcome of this study was well-being as assessed using the expanded Physician Well-being Index (ePWBI) [24]. The ePWBI consists of 7 questions that are answered by yes or no (with one point assigned for each "yes"), and 2 questions that are answered using a 7-point Likert scale (with a score of -1 to +1 assigned to each of these questions). The score was validated in a sample of 6880 US physicians, and a score of ≥3 points was associated with a higher risk of adverse outcomes, including medical error, burnout, severe fatigue, suicidal ideation and poor quality of life [24, 25]. Thus, we defined an ePWBI score of ≥3 points as poor wellbeing.

Secondary outcomes included job satisfaction, burnout, quality of life, work-life balance, mental and physical health, fatigue, depressive symptoms, and physical activity. Job satisfaction was assessed using a single-item measure rated on a 5-point Likert scale [26, 27]. Burnout was assessed using the single-item measures of depersonalisation and emotional exhaustion from the PWBI score as a measure for burnout, as done in a previous study evaluating the well-being of Swiss residents [6]. These single-item measures representing depersonalisation and emotional exhaustion are adapted from the Maslach Burnout Inventory (MBI), which is considered the gold standard for measuring burnout, but is limited by its length (i.e. 22 items) [28]. The usability of these two items as an abbreviated tool to assess burnout has been demonstrated previously [29]. Quality of life was measured by a linear analogue scale with a response range from 0 (as bad as it can be) to 10 (as good as it can be) [30, 31]. Work-life balance was assessed using the additional question of the ePWBI "My work schedule leaves me enough time for my personal/family life", rated on a 5-point Likert scale. Mental and physical health were assessed using the Medical Outcomes Study Short-Form Health Survey (SF-8). The questionnaire entails 8 items with 5- and 6-point Likert-type scales and generates norm-based T-scores ranging from 0 to 100, with higher scores indicating better health, calibrated to the general U.S. population (mean = 50, SD = 10). The sub-scale scores can be computed to two summary scores, the Physical Component Summary (PCS) and the Mental Component Summary (MCS), calculated as the weighted sum of the 8 sub-scale scores and normalized to the U.S. general population [32, 33]. Fatigue was assessed by the Stanford Sleepiness Scale, a 7-item self-assessment of one's current level of sleepiness [34]. Depressive symptoms were assessed by the 2 first items of the 9-item Patient Health Questionnaire (PHQ-2), which has been validated as a screening tool for depression [35]. The score asks the frequency of depressed mood and anhedonia, scoring each as 0 (not at all) to 3 (nearly every day). A combined score of ≥3 has been recognised as a threshold for depression [35]. Physical activity, was assessed by the daily step count, a readily available measure inversely related to health outcomes such as all-cause

mortality, type 2 diabetes, and cardiovascular events [36]. Participating physicians were asked to indicate the average number of steps taken per day for the current month, the previous month, and two months prior, according to the default health app on their smart-phone.

## Statistical analysis

We calculated that a sample size of 92 participants would provide 80% power to detect a moderate 0.5–0.6-standard deviation effect size in the primary outcome, a level that has been previously used to describe a clinically significant difference in outcomes [37, 38]. According to our knowledge, the minimally clinically important difference in the ePWBI (as well as the original PWBI) has not been described in the literature, although an ePWBI of ≥3 points was previously associated with a higher risk of adverse outcomes [24]. We estimated that a survey sample of at least 184 participants would be sufficient to meet the target sample size, assuming a response rate of at least 50% (based on a previous study in GIM in Switzerland) [6].

Characteristics of part-time and full-time clinical hospitalists were compared using chi-squared tests for categorical variables and t-tests or Wilcoxon rank sum tests for continuous variables, as appropriate. To assess the association between part-time vs. full-time clinical work and outcomes, we used propensity score methods to model the probability of the exposure (i.e. working full-time or part-time as a clinician) while accounting for potential confounders and minimizing bias [39]. Potential confounders were identified using a directed acyclic graph, and covariates were included based on review of the literature [40]. To derive the propensity scores (and thus to assess the conditional probability of working part-time given an individual participant's covariate values), [39] we developed a logistic regression model using part-time vs. full-time clinical work as the dependent variable, and age, sex, parenthood, relationship status, academic ambition, reduced work capacity due to health reasons, time since transition to a hospitalist (i.e. years of experience as a hospitalist), and number of patients to care for as the independent variables. The reference group consisted of full-time clinical hospitalists. Propensity scores were derived from the logistic regression model, and quintiles of the propensity scores were computed. The association between part-time vs. full-time clinical work and the ePWBI (as a continuous variable) was assessed using a linear regression model adjusted for quintiles of the propensity score. To analyze the ePWBI as a binary outcome with a cut-off of ≥3 to define low well-being, [41] a logistic regression model adjusted for quintiles of the propensity score was used [42]. The association between part-time vs. full-time work and secondary outcomes including job satisfaction, quality of life, fatigue, work-life balance, and mobility was similarly assessed using linear regression models, adjusting for quintiles of the propensity scores. The association between part-time vs. full time work and depressive symptoms and burnout was analyzed using a logistic regression model, adjusting for quintiles of the propensity score.

In a sensitivity analysis, we categorized hospitalists working part-time clinically, but in blocks of entire weeks, in the full-time instead of part-time group, to assess whether working in blocks may mitigate potential effects of part-time clinical work on well-being or job satisfaction. In a second sensitivity analysis, we compared hospitalists working full-time and part-time overall (instead of clinically) to assess the association between well-being and part-time overall. This sensitivity analysis was performed to further investigate whether diversification of one's work or mere reduction in working hours were driving potential differences in well-being.

Two-sided p-values of 0.05 were considered statistically significant. All statistical analyses were conducted with Stata statistical software, release 16 (Stata Corporation, College Station, TX, USA).

## Results

Of the 199 hospitalists who received the survey, 137 answered for a response rate of 69% (**Fig 1**). Thirteen participants were excluded because they met the predefined exclusion criteria, resulting in a final study sample of 124 hospitalists, of whom 57 (46%) had a full-time and 67 (54%) a part-time clinical employment (median clinical employment percentage 70%, inter-quartile range [IQR] 50–80%). Compared to full-time clinical hospitalists, those with a part-time clinical employment were older (age ≥40 years in 24% vs. 5%, p = 0.004), more likely to be female (76% vs. 54%, p = 0.011) and to have children (64% vs. 41%, p = 0.014), had a longer duration since graduation from medical school (median 8.5 years, IQR 7–11 years vs. 7 years, IQR 6–9 years, p<0.001), and more experience as a hospitalist (median 4 years, IQR 2–7 years vs. 2 years, IQR 1–5 years, p = 0.010; **Table 1**). There was no significant difference in long-term goals overall, although part-time clinicians more often wished to become general practi-tioners (25% in part-time vs. 11% in full-time clinicians), whereas full-time clinicians more often aimed to become specialists in another discipline (10% in part-time vs. 19% in full-time clinicians). The absolute amount of weekly overtime did not differ significantly between the two groups, however the proportion of overtime according to employment percentage was sig-nificantly higher for part-time than full-time clinicians (12%, IQR 8–18% vs. 8%, IQR 8–13%, p = 0.002; **Table 1**).

The most frequently reported reason for part-time clinical work among hospitalists work-ing part-time was family duties (63%), while it was more personal time (90%) followed by fam-ily duties (63%) in the 41 (72%) full time hospitalists who wished to reduce to part-time (**S1 Table**). Only 3 hospitalists (4%) in the clinical part-time group were employed in another clin-ical activity outside of the hospital, and 2 hospitalists who wished to work part-time (5%) wanted to work clinically outside of the hospital (**S1 Table**).

The mean ePWBI was lower, i.e. indicating better well-being, in part-time clinicians (mean 1.6, 95% confidence interval [CI] 0.9–2.2) compared to full-time clinicians (mean 2.2, 95% CI

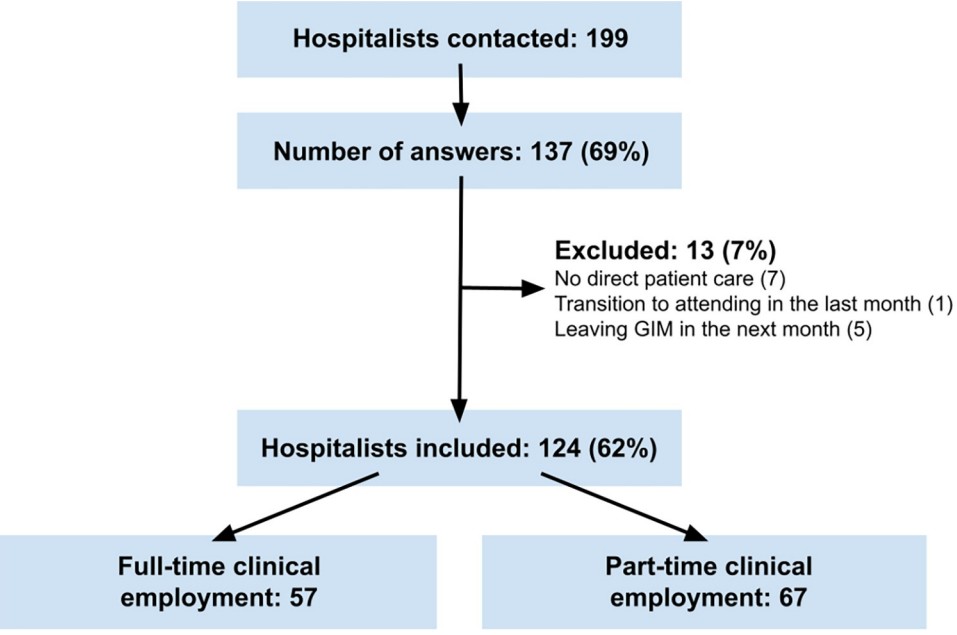

**Fig 1. Flowchart: Hospitalists of General Internal Medicine departments included in this study.** Abbreviations: GIM, general internal medicine.

**Table 1. Characteristics of participating hospitalists.**

| | Full-time clinical employment (n = 57) | Part-time clinical employment (n = 67)* | p-value |
|---|---|---|---|
| Clinical employment level in %, median (IQR) | 100 (100–100) | 70 (50–80) | < **0.001** |
| Overall employment level in %, median (IQR) † | 100 (100–100) | 80 (60–90) | < **0.001** |
| Full-time employment overall, n (%) | 57 (100%) | 14 (21%) | < **0.001** |
| Gender, n (%) | | | |
| Male | 26 (46%) | 16 (24%) | **0.011** |
| Female | 31 (54%) | 51 (76%) | |
| Age in years, n (%) | | | |
| < 40 | 54 (95%) | 51 (76%) | **0.004** |
| ≥ 40 | 3 (5%) | 16 (24%) | |
| Parenthood, n (%) | 20 (41%) | 38 (64%) | **0.014** |
| Relationship status, n (%) | | | 0.08 |
| Single | 10 (18%) | 4 (6%) | |
| Partnership | 22 (39%) | 20 (31%) | |
| Married | 25 (44%) | 37 (57%) | |
| Divorced | 0 (0%) | 2 (3%) | |
| Other | 0 (0%) | 2 (3%) | |
| Time since graduation in years, median (IQR) | 7 (6–9) | 8.5 (7–11) | < **0.001** |
| Experience as attending hospitalist in years, median (IQR) | 2 (1–5) | 4 (2–7) | **0.010** |
| Long-term goal, n (%) | | | |
| Clinical hospitalist | 30 (53%) | 31 (46%) | 0.23 |
| Academic position | 2 (4%) | 2 (3%) | |
| General practitioner | 6 (11%) | 17 (25%) | |
| Specialist in other discipline | 11 (19%) | 7 (10%) | |
| Undecided | 8 (14%) | 10 (15%) | |
| Average clinical hours per week, median (IQR) | 50 (50–55) | 38 (28–45) | < **0.001** |
| Overtime per week in hours, median (IQR) | 8 (8–13) | 7.8 (4.8–12.2) | 0.45 |
| Overtime in relation to planned working time, median (IQR) | 0.08 (0.08–0.13) | 0.12 (0.08–0.18) | **0.002** |
| Average number of night shifts per month, median (IQR) | 3 (0–4) | 1 (0–3) | **0.015** |
| Average adjusted number of night shifts per month, (median, IQR) ‡ | 3 (0–4) | 2.5 (0–5) | 0.53 |
| Average number of patients to care for per day, mean (95% CI) | 20.2 (18.7–21.8) | 18.3 (17–19.5) | 0.05 |

Abbreviations: CI: confidence interval; IQR, interquartile range. Values were missing for: number of children (n = 16); relationship status (n = 2); time since graduation (n = 1); experience as a hospitalist (n = 2); average clinical hours per week (n = 15); average number of night shifts per month (n = 4); average number of patients (n = 3).

P-values were calculated using chi-squared tests for categorical variables and t-tests or Wilcoxon rank sum tests for continuous variables.

* part-time clinical work was defined as an employment of <100% for clinical work

† refers to employment for clinical work plus any additional employment for non-clinical tasks (e.g. research)

‡ adjusted to 100% clinical employment

§ calculated by weekly overtime (in hours) divided by planned weekly working hours

1.6–2.8), although the difference was not statistically significant. Part-time clinicians were significantly less likely to have poor wellbeing (defined by an ePWBI ≥3) than full-time clinical hospitalists (31% vs. 54%, p = 0.012, **Table 2**). Among the items of the ePWBI, the largest difference between the two groups was observed for work-life balance: 72% of full-time hospitalists felt that their work schedule did not leave enough time for their personal or family life, while this was less frequently reported by part-time hospitalists (56%). In propensity-score adjusted analyses, part-time compared to full-time clinical hospitalists tended to have a lower ePWBI score (mean difference 1.00, 95% CI -2.11–0.11, p = 0.08), and they had a statistically

**Table 2. Well-being and job satisfaction among full-time and part-time hospitalists.**

| | Full-time clinical employment (n = 57) | Part-time clinical employment (n = 67) | p-value |
|---|---|---|---|
| **ePWBI score**\*, mean (SD) | 2.2 (2.2) | 1.6 (2.4) | 0.15 |
| **Poor well-being** (ePWBI ≥3)\*, n (%) | 30 (54%) | 19 (31%) | **0.012** |
| **Symptoms of burnout** in the last month § | 36 (63%) | 34 (51%) | 0.17 |
| **Job satisfaction** | | | |
| Extremely dissatisfied | 0 (0%) | 1 (2%) | 0.55 |
| Dissatisfied | 4 (7%) | 3 (5%) | |
| Neutral | 12 (22%) | 8 (13%) | |
| Satisfied | 31 (57%) | 40 (65%) | |
| Extremely satisfied | 7 (13%) | 10 (16%) | |
| **Quality of life**, mean (SD) † | 6.1 (1.7) | 6.6 (2.0) | 0.14 |
| **Work-life balance** ‡ | | | |
| Satisfactory | 8 (14%) | 20 (30%) | 0.09 |
| Unclear | 8 (14%) | 9 (14%) | |
| Not satisfactory | 41 (72%) | 37 (56%) | |
| **Mental and physical health (SF-8)** [+] | | | |
| Physical Component Summary, mean (SD) | 50.7 (7.0) | 51.6 (7.4) | 0.52 |
| Mental Component Summary, mean (SD) | 43.2 (10.3) | 47.2 (9.5) | **0.028** |
| **Fatigue**, median (IQR) ⌇ | 2 (1–3) | 1 (0–2) | 0.07 |
| **Depressive symptoms** # | 10 (18%) | 2 (3%) | **0.006** |
| **Mean daily step count** over 3 months, mean (95% CI) | 7505 (6837–8173) | 7122 (6224–8019) | 0.48 |

Abbreviations: ePWBI, extended Physician Well-Being Index; IQR, interquartile range; SD, standard deviation; SF-8: Short-form Health Survey.

All values are in n (%) unless otherwise specified. Values were missing for: ePWBI score (n = 6); job satisfaction (n = 8); quality of life (n = 5); work-life balance (n = 1); Physical Component Summary (n = 1); Mental Component Summary (n = 3); mean daily step count (n = 62).

P-values were calculated using chi-squared tests for categorical variables and t-tests or Wilcoxon rank sum tests for continuous variables.

\* the ePWBI consists of 9 questions and ranges from -2 to 9, with lower scores denoting better well-being. A score of ≥3 points was defined as poor wellbeing.

§ measured by a positive answer to at least one of the two first questions of the ePWBI

rated on a 5-point Likert scale

† measured by a linear analogue scale with a response range from 0 (as bad as can be) to 10 (as good as it can be).

‡ measured by question 9 of the ePWBI

[+] The SF-8 consists of 8 items with 5- and 6-point Likert-type scales. Each item generates a norm-based T-score ranging from 0 to 100 with higher scores indicating better health, calibrated to a mean score of 50 in the general U.S. population. The Physical and Mental Component Summary is calculated as the weighted sum of the 8 sub-scale scores and normalised to the U.S. general population.

⌇ assessed using the 7-item Stanford Sleepiness Scale, 1 being fully alert and 7 imminent sleep onset

# measured using the 2 first items of the 9-item Patient Health Questionnaire (PHQ-2).

significantly lower risk of poor well-being, with an odds ratio (OR) of 0.20 (95% CI 0.07–0.59) for an ePWBI ≥3 points (p = 0.004; **Table 3**).

There was a statistically significantly lower prevalence of depressive symptoms in part-time compared to full-time clinicians based on the PHQ-2 (3% vs. 18%; p = 0.006, **Table 2**). In the propensity-score adjusted analyses, part-time clinicians had a statistically significantly lower risk for depressive symptoms than full-time hospitalists (OR 0.14, 95% CI 0.02–0.85, p = 0.033, **Table 3**). The Mental Component Summary (MCS) score of the SF-8 indicated better mental health for part-time (mean 47.22, SD 9.54) compared to full-time physicians (mean 43.19, SD 10.30, p = 0.028), while the Physical Component Summary (PCS) did not significantly differ (**Table 2**). In the propensity-score adjusted analyses, the results were similar, with a higher mean MCS score for part-time clinicians than their full-time colleagues (mean difference 4.82, 95% CI 0.13–9.5, p = 0.044, **Table 3**) and no statistically significant difference in the PCS

**Table 3. Adjusted association between part-time vs. full-time work and outcomes based on propensity score models.**

| | Full-time clinical employment (n = 57) | Part-time clinical employment (n = 67) | p-value |
|---|---|---|---|
| ePWBI score, mean (95% CI)* | 1.95 (0.92–2.98) | 0.95 (-0.16–2.06) | 0.08 |
| Poor well-being (ePWBI ≥3), OR (95% CI)* | Ref. | 0.20 (0.07–0.59) | **0.004** |
| Symptoms of burnout in the last month, OR (95% CI) § | Ref. | 0.47 (0.18–1.22) | 0.12 |
| Job satisfaction, mean (95% CI) | 3.75 (3.38–4.11) | 3.79 (3.41–4.16) | 0.85 |
| Quality of life, mean (95% CI) † | 6.27 (5.41–7.13) | 6.73 (5.82–7.64) | 0.32 |
| Work-life balance, mean (95% CI) ‡ | 0.61 (0.24–0.98) | 0.31 (-0.07–0.71) | 0.15 |
| Physical Component Summary, mean (95% CI) + | 50.3 (47.0–53.6) | 51.0 (47.5–54.5) | 0.71 |
| Mental Component Summary, mean (95% CI) + | 44.0 (39.6–48.5) | 48.9 (44.2–53.5) | **0.044** |
| Fatigue, mean (95% CI) ¶ | 1.45 (0.86–2.04) | 0.99 (0.36–1.61) | 0.14 |
| Depressive symptoms, OR (95% CI) # | Ref. | 0.14 (0.02–0.85) | **0.033** |
| Mean daily step count, mean (95% CI) | 7962 (6883–9041) | 8710 (7400–10019) | 0.26 |

Abbreviations: CI: confidence interval; ePWBI, extended Physician Well-Being Index; IQR, interquartile range; OR, odds ratio; SF-8: Short-form Health Survey

Results were adjusted for the propensity of working part-time, as well as age, sex, parenthood, relationship status, academic ambition, reduced work capacity due to health reasons, time since transition to hospitalist, and number of patients to care for

*§†‡+¶# see Table 2

score. Among the individual items of the SF-8, the largest difference between the two groups was observed for the item evaluating the impact of emotional problems on daily functioning ("during the past 4 weeks, how much did personal or emotional problems keep you from doing your usual work, school or other daily activities?"). Symptoms of burnout, job satisfaction, overall quality of life, work-life balance, and fatigue tended to be better rated by part-time compared to full-time clinical hospitalists, although without any statistically significant differences (**Table 2**). Physical activity, as measured by the mean number of daily steps over the last 3 months did not differ between groups (7122 compared to 7505, p = 0.478), although the result was only available for 62/124 hospitalists. Similarly, the propensity-score adjusted analyses showed no statistically significant difference in the rest of the secondary outcomes (**Table 3**).

In a sensitivity analysis considering hospitalists working clinically part-time in blocks of whole weeks as full-time, results remained similar, although the beneficial association of part-time work with depressive symptoms and the MCS of the SF-8 score did not reach statistical significance (**S2 Table**). In a second sensitivity analysis comparing hospitalists working full-time and part-time overall, the association between part-time overall and better well-being, as well as a lower risk for depressive symptoms, remained statistically significant. Additionally, the positive trend towards fewer symptoms of burnout, better work-life balance and less fatigue reached a statistically significant difference in this analysis (**S3 Table**).

## Discussion

In our survey among hospitalists in GIM, half of all full-time clinicians reported poor well-being. Our results showed that part-time compared to full-time clinical work was associated with a lower risk of poor well-being. Hospitalists working part-time clinically were significantly less likely to have symptoms of depression and had a better mental health compared to their full-time colleagues. Job satisfaction, quality of life, work-life balance, and symptoms of burnout were not statistically significant between both groups.

The fact that more than half of all full-time and almost a third of part-time clinical hospitalists who completed our survey reported poor wellbeing is alarming, given that poor physician

well-being is not only associated with burnout, depression, suicidal ideation, substance abuse, and poor quality of life, [41, 43] but also negatively affects patient satisfaction and quality of care [1]. Additionally, physicians tend to underrecognize their mental and physical health issues and not seek help [43]. Overall, 40% of the physicians who answered our survey had low well-being. In other studies, low well-being ranged from 53% in resident physicians in Calgary, Canada, [44] to 18.6% in GPs in Denmark [45]. Low well-being seemed to be more frequent in board-certified hospitalists than residents (19%) in GIM in Switzerland [6]. To our knowledge, the relationship between part-time employment, both clinical and overall part-time, and well-being has not been investigated in physicians previously. Our results suggest that part-time clinical work is associated with better well-being compared to working as a full-time clinician. Potential explanations may be reduced stress and burden from work, but also increased diversification of one's tasks. We chose to study part-time clinical employment on the premise that non-clinical tasks may diversify the work of clinicians and allow them to spend time on work that is meaningful to them, increasing well-being similar to any other activity part-time physicians may choose besides their clinical work [10]. The sensitivity analysis comparing full-time and part-time employment overall also showed an association between part-time work overall and higher well-being, with a similar odds ratio to our main analysis. Overall, our analysis showed a positive association between part-time clinical work and well-being, whether the remaining time is spent on non-clinical professional tasks or tasks outside of the hospital.

The association of overall part-time work with job satisfaction was investigated in several studies. Only one, dating from 1995 and including only radiologists, found a higher job satisfaction for full-time physicians than part-time physicians, [46] while several more recent studies indicate that full-time physicians are less satisfied with their job compared to their part-time colleagues [11, 14, 15, 47]. Of two studies including specifically physicians in GIM, a higher job satisfaction in part-time workers was confirmed in a survey among internists and GPs in an ambulatory setting [47]. The other study included academic members of the Society of General Internal Medicine in the US and showed a similar job satisfaction in part-time and full-time clinicians and clinician-educators, [17] in line with our results. Overall, part-time work does not seem to have a consistent effect on job satisfaction, which may be explained by the fact that several predictors of job satisfaction are organizational variables specific to each hospital or health center environment, such as communication within the team, remuneration, feedback, or social standing of the institution the physicians work in [48].

Several previous studies showed a better work-life balance, fewer work-privacy conflicts, [11, 13, 15, 49] and a lower rate of burnout in physicians working part-time overall or their preferred number of hours compared to full-time physicians, [12, 47, 50, 51] a trend that was also observed in our study. The fact that we did not observe a statistically significant association similar to other studies may be due to our study comparing part-time clinical employment instead of part-time overall. In the sensitivity analysis using part-time overall, the association with work-life balance and burnout was statistically significant. Another reason may be insufficient power for these outcomes or the use of other questionnaires. For example, we only gathered data on symptoms of burnout rather than using the full MBI, because this test would have been too time-consuming for our survey among busy hospitalists. Alternatively, there may be a true lack of association between part-time work and work-life balance or burnout, because part-time workers are often taking care of family duties, as reported by two-thirds of part-time clinical hospitalists in our study–and are often part of dual-earner couples. Thus, they may be expected to take on a disproportionally high load of family duties during their time off clinical work, causing family role overload [52]. It is also possible that the addition of non-clinical tasks or the proportionally higher overtime of part-time physicians lead to burnout as much as in the full-time clinical employment group. While part-time work may be one potential

solution to promote work-life balance, [11, 13, 15, 49] implementation of other measures improving work-life balance of GIM physicians in a hospital setting, such as providing protected time for non-clinical tasks, prioritization of tasks and measures to improve the productivity, increasing the autonomy of physicians over their working schedule, and the development of institutional support systems such as employee assistance programs, remain important, as this may increase the attractiveness of GIM for future physicians [53]. In fact, an international systematic review of medical students' choice of specialty showed that work-life balance was significantly associated with the choice of specialty in 60% of articles, compared to 38.2% for gender, 45% for interest and 30% for prestige and income [54].

We found very few studies investigating physician's physical and mental health as a whole. One study assessing physical health did not find any association between the number of working hours and somatic complaints, which also matches our findings [55]. Concerning mental health, most studies concentrated on burnout, probably because this outcome is the most closely related to work-related outcomes, such as efficiency and medical error [5]. One study in Australia showed that junior doctors working more than 55 hours per week were more than twice as likely to report mental health issues than those working 40–44 hours per week [56]. Another study among resident physicians in Saudi Arabia showed that longer working hours were associated with higher rates of anxiety and depression, with odds of depression that were twice as high for physicians working more than 40 hours per week [57]. In a Norwegian prospective survey, the number of working hours did not influence the probability of mental health problems in first-year residents undergoing a compulsory internship year in surgery and internal medicine [58]. However, it is unclear in the article if part-time employment was a possibility during this study. Furthermore, the evaluation was based on self-assessment of overall health status rather than validated scores, which has been shown to be poor in physicians, [43] and mental health issues reported as minor were not taken into consideration.

Supporting and increasing opportunities for part-time (clinical) work may not only contribute to physician well-being and health, but could also be important to ensure retention of a healthy GIM workforce in hospitals. Of all Swiss physicians, 22% work in GIM, making it the most frequently chosen medical speciality in Switzerland [19]. However, in a hospital setting, the number of GIM specialists drops to 13% [59]. In order to prevent migration of general internist hospitalists to ambulatory care and promote their retention in the hospital setting, experts have called for increasing opportunities for part-time work in the GIM hospital setting, [7] a call that is supported by the results of our study. In addition, part-time work has been suggested to increase the attractiveness of GIM, allowing physicians to facilitate their personal development during different phases of their career [60]. A recent survey found that 68% of physicians working full-time would like to reduce to part-time employment [61]. This preference is not only the result of an increasing proportion of women (who may still choose to be mothers and have a family life) in medicine overall and in GIM, [9] but also reflects the priorities of the next generation of general internists, such as a growing equality of men and women in sharing of family duties and a better balance between work, family and spare time [62]. Our findings support the hypothesis that part-time employment or replacement of some clinical work with non-clinical professional tasks is associated with higher well-being, although it is unclear from the results of this cross-sectional survey if the association is causal. More research is needed to investigate whether part-time clinical activity of hospitalists has an impact on quality of patient care. In fact, a recent cross-sectional study among Medicare beneficiaries from the US found a higher mortality in patients treated by physicians with reduced clinical time [63].

To our knowledge, this is the first study to investigate the association between part-time work and physician well-being in physicians specialized in GIM. We found no study assessing

all aspects of well-being consecutively. The response rate of 69% was high for a physician survey, and the percentage of part-time hospitalists in our results (43% overall part-time and 54% clinical part-time hospitalists) is close to the percentage of part-time hospitalists declared by the participating GIM wards (51%). Thus, we expect the answers to the survey to be representative of our study population, although the results may not apply to each individual hospital, especially those employing only few part-time hospitalists. We could not compare characteristics of hospitalists who answered the questionnaire and those who did not, because the answers were anonymous. One limitation of our study is that our sample size is relatively small, so although it was sufficiently powered to detect significant differences in our primary outcome, it may have been underpowered concerning the secondary outcomes. The sample size was also too small for a meaningful subgroup analysis of our data (e.g. by sex). Second, although we adjusted our analysis for known confounders that influence the association between part-time work and well-being, we cannot exclude the potential for residual confounding. To ensure that participants could not be identified, we did not gather any information on the site participants were employed at, as some sites had as few as 4 potential participants. Thus, we were not able to adjust our analysis for hospital site to reflect the size of the hospital they were employed at. We did not gather data on the age of participant's children, which could further influence the workload at home and therefore affect the decision of working part-time work as well as a participants' wellbeing. We adjusted for parenthood as a potential confounder but not the number of children, because we hypothesized that having children per se rather than the number of children would causally affect whether a physician was working part-time of full-time. Third, we only looked at the situation in GIM in Switzerland, thus our result may not be generalizable to other health care systems or settings. Fourth, as this is a cross-sectional study, the association between part-time clinical employment and higher well-being shown in our results does not guarantee causation. Finally, we did not use a validated score for burnout. We included questions assessing symptoms of burnout; however, they did not include the frequency component. We decided not to include the MBI, as it would have made the survey too long and may have lowered the response rate. In this study, we did not assess patient safety or other patient outcomes.

## Conclusions

In conclusion, full-time clinical hospitalists in GIM have a high risk for poor well-being. Part-time compared to full-time clinical work is associated with better well-being and better mental health. Supporting and enabling part-time clinical work may contribute to an improvement in well-being and mental health among GIM hospitalists and their retention in hospital settings.

## Supporting information

**S1 Table. Reasons for part-time clinical work among hospitalists working part-time and among those working full-time but wishing to reduce to part-time clinical work.** Participants were able to select multiple reasons. *Other reasons listed for part-time clinical employment: administrative work, university duties.
(DOCX)

**S2 Table. Sensitivity analysis considering part-time clinicians working whole weeks (n = 22) as full-time, adjusted for quintiles of the propensity scores.** Abbreviations: CI: confidence interval; ePWBI, extended Physician Well-Being Index; IQR, interquartile range; OR, odds ratio; SF-8: Short-form Health Survey. Results were adjusted for the propensity of working part-time, as well as age, sex, parenthood, relationship status, academic ambition, reduced

work capacity due to health reasons, time since transition to hospitalist, and number of patients to care for * the ePWBI consists of 9 questions and ranges from -2 to 9, with lower scores denoting better well-being. A score of ≥3 points was defined as poor wellbeing.§ measured by a positive answer to at least one of the two first questions of the ePWBI. rated on a 5-point Likert scale. † measured by a linear analogue scale with a response range from 0 (as bad as can be) to 10 (as good as it can be). ‡ measured by question 9 of the Epwbi. [+] 8 items with 5- and 6-point Likert-type scales. Each item generates a norm-based T-score ranging from 0 to 100 with higher scores indicating better health, calibrated to a mean score of 50 in the general U.S. population. The Physical and Mental Component Summary is calculated as the weighted sum of the 8 sub-scale scores and normalised to the U.S. general population. ¶ assessed using the 7-item Stanford Sleepiness Scale, 1 being fully alert and 7 imminent sleep onset. # measured using the 2 first items of the 9-item Patient Health Questionnaire (PHQ-2). (DOCX)

**S3 Table. Sensitivity analysis considering full-time and part-time employment overall, adjusted for quintiles of the propensity scores.** Abbreviations: CI: confidence interval; ePWBI, extended Physician Well-Being Index; IQR, interquartile range; OR, odds ratio; SF-8: Short-form Health Survey. Results were adjusted for the propensity of working part-time, as well as age, sex, parenthood, relationship status, academic ambition, reduced work capacity due to health reasons, time since transition to hospitalist, and number of patients to care for * the ePWBI consists of 9 questions and ranges from -2 to 9, with lower scores denoting better well-being. A score of ≥3 points was defined as poor wellbeing. § measured by a positive answer to at least one of the two first questions of the Epwbi. rated on a 5-point Likert scale. † measured by a linear analogue scale with a response range from 0 (as bad as can be) to 10 (as good as it can be). ‡ measured by question 9 of the ePWBI. [+] 8 items with 5- and 6-point Likert-type scales. Each item generates a norm-based T-score ranging from 0 to 100 with higher scores indicating better health, calibrated to a mean score of 50 in the general U.S. population. The Physical and Mental Component Summary is calculated as the weighted sum of the 8 sub-scale scores and normalised to the U.S. general population. ¶ assessed using the 7-item Stanford Sleepiness Scale, 1 being fully alert and 7 imminent sleep onset. # measured using the 2 first items of the 9-item Patient Health Questionnaire (PHQ-2). (DOCX)

## Author Contributions

**Conceptualization:** Lisa Bretagne, Stefanie Mosimann, Christine Roten, Martin Perrig, Daniel Genné, Manfred Essig, Marco Mancinetti, Marie Méan, Pauline Darbellay Farhoumand, Lars C. Huber, Elisabeth Weber, Christoph Knoblauch, Andreas W. Schoenenberger, Sonia Frick, Eliane Wenemoser, Daniel Ernst, Michael Bodmer, Drahomir Aujesky, Christine Baumgartner.

**Data curation:** Lisa Bretagne.

**Formal analysis:** Lisa Bretagne, Christine Baumgartner.

**Funding acquisition:** Lisa Bretagne, Stefanie Mosimann, Christine Roten, Christine Baumgartner.

**Investigation:** Daniel Genné, Manfred Essig, Marco Mancinetti, Marie Méan, Pauline Darbellay Farhoumand, Lars C. Huber, Elisabeth Weber, Christoph Knoblauch, Andreas W. Schoenenberger, Sonia Frick, Eliane Wenemoser, Daniel Ernst, Michael Bodmer, Drahomir Aujesky.

**Methodology:** Lisa Bretagne, Stefanie Mosimann, Christine Roten, Martin Perrig, Christine Baumgartner.

**Project administration:** Lisa Bretagne, Christine Baumgartner.

**Supervision:** Christine Baumgartner.

**Writing – original draft:** Lisa Bretagne, Christine Baumgartner.

**Writing – review & editing:** Stefanie Mosimann, Christine Roten, Martin Perrig, Daniel Genné, Manfred Essig, Marco Mancinetti, Marie Méan, Pauline Darbellay Farhoumand, Lars C. Huber, Elisabeth Weber, Christoph Knoblauch, Andreas W. Schoenenberger, Sonia Frick, Eliane Wenemoser, Daniel Ernst, Michael Bodmer, Drahomir Aujesky.

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
