## [Decision Letter · Decision Letter 0]

4 Jul 2023

PONE-D-23-11884Association of part-time clinical work with well-being and mental health in General Internal Medicine: A survey among Swiss hospitalistsPLOS ONE

Dear Dr. Bretagne,

Thank you for submitting your manuscript to PLOS ONE. After careful consideration, we feel that it has merit but does not fully meet PLOS ONE’s publication criteria as it currently stands. Therefore, we invite you to submit a revised version of the manuscript that addresses the points raised during the review process. 

Many thanks for this interesting article dealing with a relevant and pressing issue.

I also thank the reviewers very much for their valuable input. I am sure that by following the reviewer's suggestions, the quality of the article will further increase.

Please respond to the comments of Reviewer 1. I ask you to focus mainly on points 1 (this point was also indicated by Reviewer 2), 2, 3, 4, 7. If possible also respond to points 5, 6, and 8.

Please add a short paragraph about the Swiss Health Care System in the introduction so as to help readers understand the similarities and differences with their Health Care System. Also comment on the possibility of part-time physicians then working outside the hospital, as suggested by Reviewer 4.

We look forward to receiving your revised manuscript.

Kind regards,

Lorenzo Righi

Academic Editor

PLOS ONE

Reviewers' comments:

Reviewer's Responses to Questions

**Comments to the Author**

1. Is the manuscript technically sound, and do the data support the conclusions?

Reviewer #1: Yes

Reviewer #2: Partly

Reviewer #3: Yes

Reviewer #4: Yes

2. Has the statistical analysis been performed appropriately and rigorously? 

Reviewer #1: Yes

Reviewer #2: No

Reviewer #3: Yes

Reviewer #4: Yes

3. Have the authors made all data underlying the findings in their manuscript fully available?

Reviewer #1: Yes

Reviewer #2: Yes

Reviewer #3: Yes

Reviewer #4: Yes

4. Is the manuscript presented in an intelligible fashion and written in standard English?

Reviewer #1: Yes

Reviewer #2: Yes

Reviewer #3: Yes

Reviewer #4: Yes

5. Review Comments to the Author

Reviewer #1: The authors conducted a cross-sectional online study collecting information on working hours and mental health and well-being among Swiss hospitalits. In my opinion, the topic under investigation is of great relevance for health care systems and in order to maintain sufficient patient care – not only in Switzerland and specifically in the light of physican shortage. The analysis was very detailed, including a range of important factors of well-being and health.

I have some comments and suggestions to enhance the clarity of the manuscript.

1. In your method section, you describe that physicians came from small regional hospitals or large tertiary care centres. Did you think of adding this as a confounding variable in your regression models? There is evidence that this could be important in terms of well-being, job satisfaction and mental health of physicians (DOI: 10.1055/s-0042-121596, 10.1055/a-1173-9188).

2. Sample: In my opinion, the reason why some participants were excluded is questionable. For instance, if you assume that residents are generally working full-time (by giving one reference only), you could also exclude female or older physicians, because they are more likely to work part-time as well. Besides, why did you exclude physicians planning to leave GIM in the next month (Figure 1)?

3. On page 6, line 120-123, the authors describe how they define full-time and part-time, which is very helpful (many other studies simply refer to these categories without explaining them, and it may differ between countries), however, was this based on number of hours determined by contract or does it include total hours in addition to overtime hours? It would be helpful to clarify this by one sentences.

4. On page 6, line 123-125 you mention that physicians employed full-time but with predefined percentage of work dedicated to non-clinical tasks such as research, was considered as part-time. On the other hand, in your introduction, you say that the possibility to be engaged in these non-clinical tasks – such as research - was associated with lower rates of burnout. Even if your results do not show that these activities may be linked to lower burnout rates, it may still bias your results. Even if physicians do not spend all of their time in direct patient contact, they may still be working 50hrs a week and therefore be affected by burnout. At the same time, categorizing them into the part-time group makes it sound as if “doing” research is not as valuable as working in direct patient care. Maybe the authors want to reconsider their categorization scheme or explain this categorization more comprehensively.

5. On page 8, line 8 you mention that “parenthood” was added as a covariate. Did you ask for the age or number of children? The impact of child care or the “double burden” may be different for someone with a child aged 2 or aged 17, or whether someone has oen child or four to look after. If not, this could be added to the limitation section.

6. Discussion section: It would be interesting, if the authors share their ideas regarding the causation or reason for their findings based on previous literature and give some implications. What can hospitals do with these findings? Would could be changed (in terms of working hours) in order to maintain or improve well-being?

7. In this context, it is interesting that they did not find any differences regaridng burnout. By looking at Table 1, I was wondering whether working part-time creates equal amount of stress/burnout, because neither the overtime hours nor the average number of patients to care is significantly different between the two groups. Maybe part-time physicians are stressed because they have the same amount of work but less time for it.

8. I find the amount of information ( indictaed by §,*, ♦, ‡, etc.)) below each table rather confusing – maybe the authors can find a way to avoid this accumulation, as most information is given repeatedly under each table.

Reviewer #2: The authors present a study with the aim of assessing the association between part-time compared to full-time clinical work and aspects of well-being, job satisfaction, and health among hospitalists of GIM departments in Switzerland.

The manuscript has serious limitations that prevent me from recommending publication.

Reviewer #3: The paper addresses an important issue, whether part-time compared to full-time clinical work is associated with better well-being and mental health and fewer depressive symptoms.

The paper is well written with an appropriate analytical approach and deserves publication. However, I have a comment related to selectivity bias and suggesting to address.

Out of 199 hospitalists who received the survey, 137 answered and the response rate is 69%. Thirteen participants were also excluded. I would suggest comparing 137 and 62 cases/hospitalists in terms of background characteristics and whether they differ. I hope this comparison is possible as samples were taken from hospitals.

Reviewer #4: Thanks for sending over this paper for my review, which looks at part-time and full time physicians wellbeing.

While there are some strengths and interesting features, I believe the paper can be strengthened by some further developments in terms of theoretical background and discussion.

Introduction:

Is there any theories related to having multiple jobs for physicians/ employees may contribute to wellbeing?

And under what context (eg government support, economic environment, life trajectory) this might be beneficial?

Methods:

Have the analysis separated part-time physicians with other jobs (so may potentially work full time hours)and those who only work is as a part-time physicians? If not, then I believe the interpretation of the results and thus in the discussion will need to be clear about the complex composition of the part-time group.

Data analysis:

Can you explain a bit more on the reliability or evidence that the one item measures and shorten measurements are appropriate?

Discussion:

With some part-time physicians have other job commitments but some don’t, and attributing diversified work for their higher satisfaction doesn’t sound very strong.

Line 376 -388: When quoting studies conducted in other countries, mentioning the potentially similar or different context that led to inconsistencies in results will improve the clarity and depth of discussion.

Line 389 - 401: Any other factors push and pull factors for the “trend” to go part-time and take up multiple positions and related discussions? Reflecting the interest of the “next generation” sound like an over simplification of the phenomenon.

6. PLOS authors have the option to publish the peer review history of their article (what does this mean?). If published, this will include your full peer review and any attached files.

Reviewer #1: No

Reviewer #2: No

Reviewer #3: No

Reviewer #4: No

---

## [Author Response · Author response to Decision Letter 0]

6 Aug 2023

Dear Dr. Righi,

Thank you for the opportunity to revise and resubmit our manuscript entitled “Association of part-time clinical work with well-being and mental health in General Internal Medicine: A survey among Swiss hospitalists”.

We have provided a point-by-point response to each comment below, and highlighted the changes made in the manuscript. Each author has approved the revised version, and we hope that these changes meet with your approval. 

The data analyzed for the current study are not publicly available, because this specific study was exempted from ethical approval. Additionally, the gathered data contains sensitive and personal information on hospitalists who, although anonymized, may still be identifiable. Data may be shared with researchers for reasonable scientific purposes on request if the use has been approved by an ethical committee. For data access and requests for the analytical code, researchers may contact the corresponding author.

Please do not hesitate to contact us with any remaining questions or for additional clarification. 

Thank you again for your valuable review and the opportunity to revise our manuscript. We look forward to your response.

On behalf of the investigators,

Lisa Bretagne 

Comments from the academic editor:

1. Many thanks for this interesting article dealing with a relevant and pressing issue. I also thank the reviewers very much for their valuable input. I am sure that by following the reviewer's suggestions, the quality of the article will further increase. Please respond to the comments of Reviewer 1. I ask you to focus mainly on points 1 (this point was also indicated by Reviewer 2), 2, 3, 4, 7. If possible also respond to points 5, 6, and 8.

Response: Thank you for your comment. We have now responded to all comments from Reviewer 1; see responses below. We have also addressed the comments from the other reviewers in the letter below.

2. Please add a short paragraph about the Swiss Health Care System in the introduction so as to help readers understand the similarities and differences with their Health Care System. 

Response: We have added a paragraph on the Swiss Health Care System and General Internal Medicine in Switzerland to the methods section (page 6, lines 92-105):

“We conducted an anonymized cross-sectional electronic survey among hospitalists in Switzerland. The Swiss health care system is characterized by universal access that is ensured through mandatory health care insurance.[18] Among practicing physicians, 22% are specialized in GIM, making it the most frequent specialization in Switzerland.[19] Board certification for GIM in Switzerland requires a 5-year training;[20] GIM residents can then transition to a position either as a hospitalist or an outpatient general practitioner, or may still decide to pursue a training in another medical specialty, as many subspecialties (e.g., cardiology, infectious diseases, etc.) require at least 2 years of training in GIM. Patients hospitalized on a GIM ward of a Swiss teaching hospital are cared for by a GIM resident and a supervising (board-certified) hospitalist, and other specialists are involved depending on the patient’s disease and on request of the hospitalist. The hospitalists themselves are supervised regularly (e.g., once a week) or upon request by a chief physician, who is also responsible for running the department and for other administrative tasks, teaching, and research. Overall, the hospitalists have the overview and bear the ultimate responsibility for optimal patient care.

The survey was sent to hospitalists from GIM departments of 14 Swiss hospitals, […]”

We had the impression that the description of the Swiss Health Care System fits best in the first paragraph of the methods section. If you think that this should be moved to the introduction section, please let us know. 

3. Also comment on the possibility of part-time physicians then working outside the hospital, as suggested by Reviewer 4.

Response: Only 3 hospitalists (4% of all part-time hospitalists) had another clinical activity outside of the hospital (S1 Table). We do not think that this number is high enough to have any major influence on our results. Additionally, we did not find any previous literature on having multiple part-time jobs outside the hospital and well-being. We have now commented on this in the results section (page 13, lines 248-251):

“Only 3 hospitalists (4%) in the clinical part-time group were employed in another clinical activity outside of the hospital, and 2 hospitalists who wished to work part-time (5%) wanted to work clinically outside of the hospital (S1 Table).”

Reviewer 1:

The authors conducted a cross-sectional online study collecting information on working hours and mental health and well-being among Swiss hospitalists. In my opinion, the topic under investigation is of great relevance for health care systems and in order to maintain sufficient patient care – not only in Switzerland and specifically in the light of physician shortage. The analysis was very detailed, including a range of important factors of well-being and health.

Response: Thank you for your comment.

I have some comments and suggestions to enhance the clarity of the manuscript.

1. In your method section, you describe that physicians came from small regional hospitals or large tertiary care centres. Did you think of adding this as a confounding variable in your regression models? There is evidence that this could be important in terms of well-being, job satisfaction and mental health of physicians (DOI: 10.1055/s-0042-121596, 10.1055/a-1173-9188).

Response: Thank you for raising this important point. Unfortunately, we had to remove the variable “site” from our questionnaire to participants because of confidentiality concerns, as some centres only had few possible participants and they would have been immediately identifiable. Thus, we could not adjust our analysis for the site the participants worked at. We agree that this could be a confounding factor and we have added this to the limitations (page 20, lines 449-452):

“To ensure that participants could not be identified, we did not gather any information on the site participants were employed at, as some sites had as few as 4 potential participants. Thus, we were not able adjust our analysis for hospital site to reflect the size of the hospital they were employed at.” 

2. Sample: In my opinion, the reason why some participants were excluded is questionable. For instance, if you assume that residents are generally working full-time (by giving one reference only), you could also exclude female or older physicians, because they are more likely to work part-time as well. Besides, why did you exclude physicians planning to leave GIM in the next month (Figure 1)?

Response: We tried to get a sample that would be representative of the typical (Swiss) physician in GIM working in a hospital setting (i.e., hospitalist). First, we did not include resident physicians because in Switzerland, several years of residency in GIM is mandatory for many other specialties (e.g., cardiology, infectious diseases, endocrinology, etc.). Because of this, many residents in GIM do not envision a career in GIM. In addition, residents who want to become family doctors also need to complete at least 2 years in GIM in a hospital setting. In our opinion, residents in GIM do not reflect the future in-hospital GIM workforce, which is why we did not include them in our study. Additionally, very few hospitals in Switzerland allow part-time work for residents, which would have made the sampling difficult.

Second, we excluded hospitalists who did not have any patient contact because the aim of the study was to assess the association between the amount of clinical work and well-being. Third, we also excluded hospitalists who had transitioned in the last month as the transition is usually perceived as stressful and they may not have had enough time to adapt to their new position, so their responses concerning well-being may not be representative of hospitalist workforce. Lastly, we excluded physicians who were leaving GIM in the next month, as they were transitioning to another specialty and thus, in our opinion, they were similarly not representative of the hospitalist workforce. We now extended the rationale for excluding these participants in our methods section (page 7, lines 106-116):

“We deliberately excluded resident physicians from the survey as a considerable part of the residents working in GIM departments will switch to speciality training.[21] Furthermore, few hospitals in Switzerland allow part-time employment of residents in GIM, thus only few residents work part-time in GIM,[6] and it would have been difficult to get a big enough sample size of participants for meaningful results. By focusing on hospitalists, we intended for the study population to be more homogeneous and representative of the “general internist”. We excluded chief-physicians in hospital medicine, hospitalists without direct patient care on a ward, those who had newly transitioned to a position as hospitalist within the last month or those who intended to leave GIM within the next month, as their answers may not be representative of the GIM hospitalist workforce.”

3. On page 6, line 120-123, the authors describe how they define full-time and part-time, which is very helpful (many other studies simply refer to these categories without explaining them, and it may differ between countries), however, was this based on number of hours determined by contract or does it include total hours in addition to overtime hours? It would be helpful to clarify this by one sentences.

Response: The categories for full-time and part-time were based on the number of hours defined in the hospitalists’ contract. Overtime was presented for each group in the baseline characteristics (Table 1).

We now clarified this in the methods section (page 8, line 137-138):

“The percentage of clinical employment was based on self-report of the number of hours specified in hospitalists’ contracts, and did not include overtime.”

4. On page 6, line 123-125 you mention that physicians employed full-time but with predefined percentage of work dedicated to non-clinical tasks such as research, was considered as part-time. On the other hand, in your introduction, you say that the possibility to be engaged in these non-clinical tasks – such as research - was associated with lower rates of burnout. Even if your results do not show that these activities may be linked to lower burnout rates, it may still bias your results. Even if physicians do not spend all of their time in direct patient contact, they may still be working 50hrs a week and therefore be affected by burnout. At the same time, categorizing them into the part-time group makes it sound as if “doing” research is not as valuable as working in direct patient care. Maybe the authors want to reconsider their categorization scheme or explain this categorization more comprehensively.

Response: Our hypothesis was that non-clinical activities, such as research and teaching, combined with clinical work contribute to a better well-being compared to a full-time clinical activity (see our reference 10), given that this results in a diversification of one’s work/everyday life, similar to working part-time and thus spending time for e.g. childcare. This is why we decided to compare part-time and full-time clinical employment rather than overall employment in our main analysis. We now explained this categorization more comprehensively (page 8, lines 133-137):

“Physicians employed full time, but with a predefined percentage of their work specifically dedicated to non-clinical tasks (i.e. research, teaching, or administration), were considered as part-time clinicians, because we hypothesized that diversification of one’s work and engaging in meaningful non-clinical tasks may affect well-being similar to engaging in meaningful tasks outside of the hospital during time off.[10]”

However, we agree with the reviewer that this may be debatable, and that overall working hours rather than diversification of one’s work may contribute to reduced well-being. Therefore, we performed a sensitivity analysis comparing part-time and full-time employment overall (i.e., with inclusion of employment for non-clinical tasks). We now further clarified this in the methods section (page 11, lines 208-212):

“In a second sensitivity analysis, we compared hospitalists working full-time and part-time overall (instead of clinically) to assess the association between well-being and part-time overall. This sensitivity analysis was performed to further investigate whether diversification of one’s work or mere reduction in working hours were driving potential differences in well-being.”

The results of this sensitivity analysis similarly showed a statistically significant difference for well-being, similar to the main results, as well as reduced symptoms of burnout, improved work-life balance, reduced fatigue and symptoms of depression (see S3 Table). In our main analysis, well-being, depressive symptoms and mental health showed a statistical difference between the groups. 

It is possible and even probable that research may lead to more overtime; however, we wanted to show that even if hospitalists do not want to or cannot reduce overall working time, replacing some clinical time with other non-clinical activities may be way of improving well-being, as explained in the discussion section (page 17, lines 354-364):

“We chose to study part-time clinical employment on the premise that non-clinical tasks may diversify the work of clinicians and allow them to spend time on work that is meaningful to them, increasing well-being similar to any other activity part-time physicians may choose besides their clinical work. [10] The sensitivity analysis comparing full-time and part-time employment overall also showed an association between part-time work overall and higher well-being, with a similar odds ratio to our main analysis. Overall, our analysis showed a positive association between part-time clinical work and well-being, whether the remaining time is spent on non-clinical professional tasks or tasks outside of the hospital.

5. On page 8, line 8 you mention that “parenthood” was added as a covariate. Did you ask for the age or number of children? The impact of child care or the “double burden” may be different for someone with a child aged 2 or aged 17, or whether someone has one child or four to look after. If not, this could be added to the limitation section.

Response: We asked for the number of children but not their age. We decided to adjust for parenthood as a potential confounder for the relationship between level of employment and wellbeing and not the number of children, because we hypothesized that having children per se rather than the number of children would causally affect whether someone was working full-time of part-time. We have added this in the limitations (pages 20-21, lines 452-457):

“We did not gather data on the age of participant’s children, which could further influence the workload at home and therefore affect the decision of working part-time work as well as a participants’ wellbeing. We adjusted for parenthood as a potential confounder but not the number of children, because we hypothesized that having children per se rather than the number of children would causally affect whether a physician was working part-time of full-time.”

6. Discussion section: It would be interesting, if the authors share their ideas regarding the causation or reason for their findings based on previous literature and give some implications. What can hospitals do with these findings? Would could be changed (in terms of working hours) in order to maintain or improve well-being?

Response: As mentioned in the discussion (page 17, lines 350-351), we have found no previous studies investigating employment level of physicians and well-being overall, although previous literature on single components of well-being showed a better work-life balance, fewer work-privacy conflicts, and a lower rate of burnout and depression in physicians working part-time overall or their preferred number of hours compared to full-time physicians (pages 17-19, lines 365-415). Our results suggest that reducing work hours or replacing some clinical time with non-clinical time may lead to better well-being in GIM hospitalists. Potential reasons for improved well-being may be reduced stress and burden from work, but also a diversification on one’s tasks in everyday life. We have now added this to the discussion (page 17, lines 353-354).

“Potential explanations may be reduced stress and burden from work, but also increased diversification of one’s tasks.”

However, the causality this association between well-being and part-time work cannot be shown in this study using cross-sectional survey data. 

A recent cross-sectional study among a random sample of Medicare beneficiaries in the USA (Kato et al., JAMA Intern Med 2021, see our new reference 63) has shown a higher mortality in patients cared for by hospitalists with reduced clinical time. We think additional studies are necessary before any recommendation can be made for institutions to promote part-time work. We have now extended the discussion section accordingly (page 20, lines 429-435):

“Our findings support the hypothesis that part-time employment or replacement of some clinical work with non-clinical professional tasks is associated with higher well-being, although it is unclear from the results of this cross-sectional survey if the association is causal. More research is needed to investigate whether part-time clinical activity of hospitalists has an impact on quality of patient care. In fact, a recent cross-sectional study among Medicare beneficiaries from the US found a higher mortality in patients treated by physicians with reduced clinical time.[63]”

7. In this context, it is interesting that they did not find any differences regarding burnout. By looking at Table 1, I was wondering whether working part-time creates equal amount of stress/burnout, because neither the overtime hours nor the average number of patients to care is significantly different between the two groups. Maybe part-time physicians are stressed because they have the same amount of work but less time for it.

Response: That is a valid hypothesis and has been added to the discussion (page 18, lines 390-392):

“It is also possible that the addition of non-clinical tasks or the proportionally higher overtime of part-time physicians lead to burnout as much as in the full-time clinical employment group.”

8. I find the amount of information ( indicated by §,*, ♦, ‡, etc.)) below each table rather confusing – maybe the authors can find a way to avoid this accumulation, as most information is given repeatedly under each table.

Response: we have removed the footnotes that were repeatedly given and only left it under Table 2 (see footnotes of Table 3).

Reviewer 2:

The authors present a study with the aim of assessing the association between part-time compared to full-time clinical work and aspects of well-being, job satisfaction, and health among hospitalists of GIM departments in Switzerland.

The manuscript has serious limitations that prevent me from recommending publication.

Response: We thank the reviewer for his review and would be happy to receive more detailed information on how he came to this decision, so that we can give our best to improve the manuscript.

Reviewer 3:

The paper addresses an important issue, whether part-time compared to full-time clinical work is associated with better well-being and mental health and fewer depressive symptoms.

The paper is well written with an appropriate analytical approach and deserves publication. However, I have a comment related to selectivity bias and suggesting to address.

Response: thank you for your comment.

1. Out of 199 hospitalists who received the survey, 137 answered and the response rate is 69%. Thirteen participants were also excluded. I would suggest comparing 137 and 62 cases/hospitalists in terms of background characteristics and whether they differ. I hope this comparison is possible as samples were taken from hospitals.

Response: Unfortunately, we do not have any information on the hospitalists who received the survey but did not fill out the questionnaire as the answers are anonymous. Therefore, we do not know who answered and who did not and thus we cannot compare their characteristics. We have added this to the limitations (page 20, lines 443-444):

“We could not compare characteristics of hospitalists who answered the questionnaire and those who did not, because the answers were anonymous.”

Reviewer 4:

Thanks for sending over this paper for my review, which looks at part-time and full time physicians wellbeing.

While there are some strengths and interesting features, I believe the paper can be strengthened by some further developments in terms of theoretical background and discussion.

Response: thank you for your comment and your review.

1. Introduction: Is there any theories related to having multiple jobs for physicians/ employees may contribute to wellbeing? And under what context (e.g. government support, economic environment, life trajectory) this might be beneficial?

Response: We found no literature on the impact of working in multiple part-time jobs on well-being in literature, neither for physicians nor other employees. While working in several clinical jobs is frequent among certain specialists in Switzerland (e.g., working 3 days in hospital and 2 days in a private practice), it is not as frequent in GIM and is more frequently done by chief physicians than hospitalists included in our survey. In our study, only 3 hospitalists worked another clinical job outside of the hospital (4% of part-time hospitalists, see S1 Table). We have added this information in the results section (page 13, lines 248-251):

“Only 3 hospitalists (4%) in the clinical part-time group were employed in another clinical activity outside of the hospital, and 2 hospitalists who wished to work part-time (5%) wanted to work clinically outside of the hospital (S1 Table).”

2. Methods: Have the analysis separated part-time physicians with other jobs (so may potentially work full time hours) and those who only work is as a part-time physicians? If not, then I believe the interpretation of the results and thus in the discussion will need to be clear about the complex composition of the part-time group.

Response: Yes, we separated physicians working part-time with another professional activity next to clinical work and those without. In the main analysis, we only looked at part-time clinical work, independently on whether they had another professional activity or not. In a sensitivity analysis, we separated the sample in a group of participants working part-time overall or full-time overall, taking into account any other professional activity physicians may have. We now clarified this in the methods section (please also see our answer to comment 4 from reviewer 1). We now also made this clearer in the discussion on page 17, lines 354-358:

“We chose to study part-time clinical employment on the premise that non-clinical tasks may diversify the work of clinicians and allow them to spend time on work that is meaningful to them, increasing well-being as any other activity part-time physicians may choose besides their clinical work.[10]”

and on page 20, lines 429-431:

Our findings support the hypothesis that part-time employment or replacement of some clinical work with non-clinical professional tasks is associated with higher well-being, […]

3. Data analysis: Can you explain a bit more on the reliability or evidence that the one item measures and shorten measurements are appropriate?

Response: We have cited several studies in the manuscript on the validity of our outcome measures including the one-item measures and the shortened measurements used (see our references 26-36 in the manuscript). We decided to use one-item measures and shortened measurements to keep the questionnaire length to a minimum, thus encouraging participation.

4. Discussion: With some part-time physicians have other job commitments but some don’t, and attributing diversified work for their higher satisfaction doesn’t sound very strong.

Response: Our premise was that working part-time as a clinician with other professional activities may be beneficial on well-being, similar to working part-time without any other professional activity (but potentially other commitment such as childcare), as this enables physicians to spend time on work that is most meaningful to them (see our reference 10: Career fit and burnout among academic faculty, Shanafelt et al., 2009). That is why we decided to separate participants to groups with part-time clinical work or full-time clinical work. The higher well-being in the part-time clinical group could be due to any activity physicians chose to do besides their clinical work, be it a non-clinical professional activity, physical activity, family or social time. We have clarified this is the discussion (see our response to your comment 2).

5. Line 376 -388: When quoting studies conducted in other countries, mentioning the potentially similar or different context that led to inconsistencies in results will improve the clarity and depth of discussion.

Response: We have added more context to the study showing inconsistent results with our study (page 19, lines 410-415)

“In a Norwegian prospective survey, the number of working hours did not influence the probability of mental health problems in first-year residents undergoing a compulsory internship year in surgery and internal medicine.[58] However, it is unclear in the article if part-time employment was a possibility during this study. Furthermore, the evaluation was based on self-assessment of overall health status rather than validated scores, which has been shown to be poor in physicians,[43] and mental health issues reported as minor were not taken into consideration.”

6. Line 389 - 401: Any other factors push and pull factors for the “trend” to go part-time and take up multiple positions and related discussions? Reflecting the interest of the “next generation” sound like an over simplification of the phenomenon.

Response: The high rates of burnout, combined with a high number of female physicians and a trend towards equal sharing of family duties between men and women seem to be the main reason cited in articles. Physicians, general internists included, are not ready to sacrifice their hobbies and family life to their job anymore (see our reference 62: Promising Future in General Internal Medicine for the Next Generation of Physicians. Allenbach et al. 2018). We now specified this in the discussion section (pages 19-20, lines 425-429):

“This preference is not only the result of an increasing proportion of women (who may still choose to be mothers and have a family life) in medicine overall and in GIM,[9] but also reflects the priorities of the next generation of general internists, such as a growing equality of men and women in sharing of family duties and a better balance between work, family and spare time.[62] ”

---

## [Editor Report · Decision Letter 1]

8 Aug 2023

Association of part-time clinical work with well-being and mental health in General Internal Medicine: A survey among Swiss hospitalists

PONE-D-23-11884R1

Dear Dr. Bretagne,

It is not common to move an article directly from "major revision" to "accepted" but in this case the work done by the authors was nothing less than excellent. So, I am and we are pleased to inform you that your manuscript has been judged scientifically suitable for publication and will be formally accepted for publication once it meets all outstanding technical requirements.

Kind regards,

Lorenzo Righi

Academic Editor

PLOS ONE
---

## [Editor Report · Acceptance letter]

19 Sep 2023

PONE-D-23-11884R1 

Association of part-time clinical work with well-being and mental health in General Internal Medicine: a survey among Swiss hospitalists 

Dear Dr. Bretagne:

I'm pleased to inform you that your manuscript has been deemed suitable for publication in PLOS ONE. Congratulations! Your manuscript is now with our production department. 

Kind regards, 

on behalf of

Dr. Lorenzo Righi 

Academic Editor

PLOS ONE